# AL Amyloidosis and Multiple Myeloma: A Complex Scenario in Which Cardiac Involvement Remains the Key Prognostic Factor

**DOI:** 10.3390/life13071518

**Published:** 2023-07-06

**Authors:** Rafael Ríos-Tamayo, Isabel Krsnik, Manuel Gómez-Bueno, Pablo Garcia-Pavia, Javier Segovia-Cubero, Ana Huerta, Clara Salas, Ramona Ángeles Silvestre, Amelia Sánchez, Marta Manso, Laura Delgado, Juan José Lahuerta, Joaquín Martínez-López, Rafael F. Duarte

**Affiliations:** 1Hospital Universitario Puerta de Hierro, IDIPHISA, CIBERCV, 28222 Majadahonda, Spain; isabel.krsnik@salud.madrid.org (I.K.); mgomezbueno@secardiologia.es (M.G.-B.); pablogpavia@yahoo.es (P.G.-P.); jsecu@jsecu.es (J.S.-C.); ana.huerta@me.com (A.H.); klarasalas@yahoo.es (C.S.); monsilvest@gmail.com (R.Á.S.); asguerrero@salud.madrid.org (A.S.); marta.manso@salud.madrid.org (M.M.); laura.delgado@salud.madrid.org (L.D.); rduarte.work@gmail.com (R.F.D.); 2Hospital Universitario 12 de Octubre, Instituto de Investigación del Hospital Universitario 12 de Octubre, 28041 Madrid, Spain; jjlahuerta@telefonica.net (J.J.L.); jmarti01@med.ucm.es (J.M.-L.)

**Keywords:** AL amyloidosis, multiple myeloma, comorbidity, prognosis, cardiac amyloidosis

## Abstract

Monoclonal gammopathies (MGs) are a wide range of diseases that may evolve or progress over time. Comorbidity plays a critical role in this setting. The co-occurrence of two MGs is not a rare event. The evidence on the association of systemic light chain (AL) amyloidosis and multiple myeloma (MM) is scarce and controversial. Herein we aim to address this topic in a large series of patients of a referral center. All consecutive AL amyloidosis patients treated at our center from January 2005 to April 2023 were prospectively enrolled in a clinical and epidemiological registry. 141 patients diagnosed with AL amyloidosis were included, of which 7 (5%) had localized whereas 134 presented with systemic disease. The heart was the most frequently affected organ (90.3%). 25 patients (18.7%) fulfilled the IMWG diagnostic criteria of MM (AL/MM). Time-dependent association between AL and MM showed that the synchronous pattern is more frequent than the appearance of a second primary malignancy. The diagnostic delay was six months (m). Patients with AL/MM had a poorer median overall survival (OS) than AL-only patients (35.5 m, CI 95% 0–88.9, vs. 52.6 m, CI 95% 16.7–88.5), but this difference was not statistically significant. The prognosis in AL is dominated by the heart involvement, which is massive in this series. In our Cox regression model, only three prognostic variables remain as independent prognostic factors: age, N-terminal pro-brain natriuretic peptide (≥8500 ng/L), and undergoing an autologous stem cell transplant, whereas left ventricular ejection fraction shows a marginal effect. More and large studies focusing on the AL/MM association are needed to uncover the characteristics and prognostic impact of this association.

## 1. Introduction

Monoclonal gammopathies (MGs) encompass a heterogeneous group of disorders characterized by the presence of a monoclonal protein (M-protein) in blood and/or urine. This abnormal protein can be a complete immunoglobulin (Ig) or only a heavy (HC) or light chain (LC), produced by a B-cell or plasma cell (PC) clone, which is usually located in the bone marrow (BM) [1].

Systemic light chain (AL) amyloidosis and multiple myeloma (MM) are classified within the category of “Plasma cell neoplasms (PCN) and other diseases with paraproteins” in the fifth edition of the World Health Organization (WHO) classification of lymphoid tumors (WHO-HAEM5) [2]. AL amyloidosis is included in the family of “Diseases with monoclonal Ig deposition” whereas MM is incorporated into the family of “PCNs”.

AL amyloidosis is a rare and challenging disease, characterized by the presence of an amyloid-related syndrome in patients with a PC and/or a B-cell clone in the bone marrow (BM). The abnormal PCs produce LC that get converted to amyloid fibrils, which tend to be deposited in various tissues, finally causing organ failure. The evidence of AL-related amyloid by biopsy is mandatory. On the other hand, MM is a complex and heterogeneous entity defined by the demonstration of a biopsy-proven plasmacytoma or a BMPC clone ≥ 10% along with at least one myeloma defining event (evidence of end organ damage or a biomarker of malignancy). A serum or urine M-protein can also be shown in most cases. Both entities have similarities and differences, and both diseases can be shown in the same patient at the same time or at different times. Therefore, three scenarios can be anticipated (Figure 1):A.Patients (pts) diagnosed with AL that develop MM in the future as a second primary malignancy.B.Pts diagnosed with AL and MM at the same time, synchronously.C.Pts diagnosed with MM in which AL appears later as a second primary malignancy.

The first scenario (A) is very infrequent. In a series of 1596 pts with primary systemic AL amyloidosis (Mayo Clinic, 1960–1994), only 6 of them had delayed progression to MM, at least six months (m) after the diagnosis of AL [3]. The last scenario (C) is also very rare. In another study from Mayo Clinic with 4319 MM pts (1990–2008) with at least 6 m of follow-up, the authors identified 47 (1.1%) pts in whom the diagnosis of AL followed the one of MM by at least 6 m [4]. However, in a recent real-world population-based registry in Sweden, in a series of 846 AL pts, 10% had a history of MM [5]. Remarkably, this study also shows that incidence of AL is increasing, reaching 15.1 cases per million in 2019.

Scenario B is probably the most frequent in daily clinical practice. However, the demonstration of both diseases at the same time is commonly used as a mutual exclusion criterion in clinical trials. Therefore, the available evidence comes largely from clinical case reports and some real-world series with heterogeneous results. Clinicians should be aware that comorbidity plays a critical prognostic role. Unfortunately, the comorbidity assessment is far from being standardized in a real-life setting, and a comprehensive approach is needed to avoid underreporting key comorbidities. This is the case for AL and MM. When one of these diagnostics is first confirmed, an additional effort should be made to rule out the other disease, keeping in mind the difficulty to carry out an efficient diagnostic workup in sometimes complex clinical backgrounds, and the need to fulfill standard diagnostic criteria. The most relevant studies on the synchronous association of MM and AL are shown in Table 1, either considering AL [6,7,8,9,10,11] or MM [12,13,14,15,16] as the primary disease, including smoldering MM (SMM) in some of them.

We, therefore, studied the association of MM in a large series of AL in a referral center with a multidisciplinary unit dedicated to AL and MM pts, located in Madrid, Spain. Given the heterogeneous results in the literature and the controversy on this topic, our study aims to shed light on the frequency, pattern and characteristics of patients with the co-occurrence of both entities.

## 2. Materials and Methods

### 2.1. Patients

All consecutive AL pts diagnosed with AL and treated at our center were prospectively included in a local registry from January 2005 to April 2023. Our hospital became a referral center during the last decade, attending and treating pts from all Spanish regions, particularly those with cardiac involvement.

Since 2014, the diagnosis of AL and MM was made according to standard diagnostic criteria of the International Myeloma Working Group (IMWG) [17]. For the aim of this work, pts diagnosed before 2014 were also retrospectively assessed by the same criteria.

### 2.2. Variables

Both clinical and laboratory variables of prognostic interest were prospectively collected. At the moment of diagnosis, the following variables were recorded: age, sex, race, Eastern Cooperative Oncology Group (ECOG) performance status score, the occurrence of weight loss, body mass index (Kg/m^2^), number of comorbidities, number of biopsies, number of organs clinically involved, diagnostic delay, therapeutic delay, presence of previously documented monoclonal gammopathy of uncertain significance (MGUS) or carpal tunnel syndrome. Laboratory variables analyzed were as follows: complete blood count, basic biochemistry, albumin, β2-microglobulin, C-reactive protein (CRP), lactate dehydrogenase (LDH), serum creatinine (Cr), type of M-protein, level of M-protein in serum, Igs dosage, free light chain (FLC) ratio (FLCr, involved/uninvolved) and FLC difference (FLCd), 24-h urine proteinuria, bone marrow plasma cells (BMPC, %), N-terminal pro-brain natriuretic peptide (NT-proBNP), cardiac troponin I (cTnI), thickness of the interventricular septum and the posterior wall of the left ventricle, and left ventricular ejection fraction (LVEF). Classical immunoparesis (CIP) was documented when any polyclonal Ig had a level below the lower limit of the normal range. Comorbidities were analyzed as previously described [7]. Imaging tests changed over time: conventional skeletal radiography or magnetic resonance imaging were used in the early years whereas ¹⁸F-fluorodeoxyglucose (FDG) positron emission tomography/computed tomography (PET/CT) was preferably used in recent years.

### 2.3. End-Points

Diagnostic delay was measured in months (m) from the date of the first related symptom to the date of the diagnostic biopsy. Therapeutic delay was calculated in days (d) from the date of the diagnostic biopsy to the first day of treatment. The cases diagnosed in other centers were confirmed in the original diagnostic biopsy whenever possible by our pathologist and a new diagnostic report was issued. Overall survival (OS) was measured in m from the moment of the diagnosis to the death for any cause.

### 2.4. Statistics

Comparisons for categorical variables among different groups were made with the χ^2^-test, using Fisher’s exact test when appropriate. Comparisons of means of quantitative continuous variables between two groups were made with the *t*-test. OS curves were estimated using the Kaplan–Meier method, and comparisons among groups were carried out with the log-rank test. Cox proportional hazards were used for the calculation of hazard ratios (HR) for each variable. For multivariate analysis, factors with prognostic significance at 0.05 level were introduced into a Cox proportional hazards model (backward analysis). All *p*-values were two-sided. No imputation for missing data was used. Data were analyzed with SPSS v20 software.

## 3. Results

### 3.1. Localized versus Systemic AL Amyloidosis

141 pts have been prospectively enrolled in the registry, of which 7 (5%) had localized AL: skin (1), gastrointestinal (1), lung (5). Figure 2 shows the striking difference between pts with localized vs. systemic AL in terms of OS; this is the reason why this subgroup of pts has been eliminated from the main analysis.

### 3.2. Systemic AL Amyloidosis

134 pts had systemic AL, being the basis of the study. The median age at diagnosis was 64.5 years (IQR 55–72.2) and 72 (53.7%) were men. The median diagnostic delay was 6 m (IQR 4–12) and the median therapeutic delay was 19 d (IQR 9.5–34.5). The ECOG distribution (%) was: 0 (0.7), 1 (20.1), 2 (41.8), 3 (32.8), and 4 (4.5). 43 pts (32.1%) presented baseline weight loss. The most frequent type of M-protein in serum was FLC only (49.6%), followed by IgG (24.1%), IgA (11.3%), Bi-clonal (9.8%), IgD (2.3%), IgM (1.5%) and non-secretory (1.5%). Median 24-h proteinuria was 0.59 g (IQR 0.18–4.1) and Bence-Jones was positive in 77.5%. Globally, the type of FLC was lambda in most cases (86.7%). CIP was documented in 80%. The median number of involved organs was 2.5 (IQR 2–3). The heart was the key organ involved (90.3%), followed by the kidney (45.5%) and BM (40.3%).

### 3.3. AL Amyloidosis with Concurrent MM

25 pts (18.7%) fulfilled IMWG criteria of MM, 2 previous SMM (scenario C) and 23 synchronous MM (scenario B). Therefore, the cohort was divided into two groups: MM/AL and AL. The MM/AL group demonstrated significantly higher values of FLCr, FLCd, and BMPC; besides, lower values of serum creatinine were also found. However, the percentage of heart involvement and other cardiac variables were similar in both groups. Table 2 shows the characteristics of key prognostic variables. Median OS of the full cohort was 45.9 m (95% CI, 18.4–73.4) (Figure 3A). Regarding results over time, two periods were analyzed, 2005–2013 and 2014–2023. Median OS was 33.7 m, CI 95% 16.3–51.1 and 57.6 m, CI 95% 31.4–83.8, *p* = 0.16, respectively.

Pts with MM/AL had a median OS shorter than the AL group (35.5 m, CI 95% 0–88.9, vs. 52.6 m, CI 95% 16.7–88.5), but the difference was not statistically significant (Figure 3B). The prognostic impact of cardiac biomarkers, particularly NT-proBNP is well known. In our series, pts with a baseline NT-proBNP ≥ 8500 pg/mL (n = 99) exhibited a poor median OS compared with those pts with lower values, 6.7 m, CI 95% 0–16.4 vs. 79.2 m, CI 95% 43.8–114.6, *p* < 0.000 (Figure 3C). A similar result occurred with pts with LVEF ≥ 50% vs. those with lower values (n = 30), 68.2 m, CI 95% 36.1–100.3 vs. 11 m, CI 95% 0.8–21.2, *p* = 0.08. The subgroup of pts without heart involvement is very low in our series (n = 13), and, as expected, they showed a remarkable median OS of 122.9 m, CI 95% 38.7–207.1. On the other hand, the median OS of patients who underwent an autologous stem cell transplant (n = 29, 21.6%) is not still reached vs. 31.9 m, CI 95% 12.4–51.4, for those who are not candidates by now (*p* < 0.000) (Figure 3D).

All prognostic variables statistically significant in the univariate analysis are shown in Table 3, besides the effect of MM for reference. Finally, the three key independent prognostic variables for OS in the Cox regression model were: age, NT-proBNP ≥ 8500, and ASCT, whereas LVEF has a marginal effect.

## 4. Discussion

MM and AL are two related but independent entities that cause damage and failure in key target organs by different pathogenetic mechanisms. Their estimated incidence is variable around the world. Broadly, the incidence of MM is about five times higher than that of AL, e.g., in Spain, the European-age-standardized incidence rate in the last decade was about 5/100,000 person-years and the crude estimated crude incidence in 2021 was 6.7 [18,19], whereas the estimated crude incidence of AL in 2018 was 1.19/100,000 person-years [20]. According to most registries, there is a growing trend showing a global increase in the incidence of both entities [5,18]. Despite remarkable progress in the last decade, both diseases are still considered incurable. Their respective OS is progressively increasing in most population-based registries, but the gap with age-matched normal populations is still striking.

Regarding the kind of association between MM and AL, AL may develop as a secondary primary neoplasm after MM (scenario C), and on the contrary, MM as a secondary primary neoplasm after AL (scenario A) can also occur, although this is a rarer event. The synchronous diagnosis of both diseases (scenario B) is relatively common in clinical practice. Probably, the most appropriate period to define synchronous cancers is 4 m after the diagnosis of the first cancer [21]. Due to the relative incidence of MM and AL, it is usually more common to have a diagnosis of MM first and confirm AL later with the corresponding biopsy. AL is considered an underreported entity. This may be due in part to a lack of clinical suspicion when facing pts with unspecific symptoms. In the setting of a newly diagnosed MM (NDMM), the possibility of an associated AL is not always considered. Considering that the heart is the organ more commonly involved, it could be appropriate to have cardiac biomarkers (NT-proBNP and troponin) as well as an electrocardiogram and echocardiogram in every NDMM at baseline. Moreover, the convenience of a potential kidney biopsy should be commented on with the nephrologist, in case that BM and subcutaneous fat biopsies are negative.

In our series, we did not find any case of MM as secondary primary neoplasm (scenario A), only two cases (1.5%) had SMM as the primary disease before the appearance of AL (scenario C) and 23 cases (17.2%) presented with synchronous MM/AL (scenario B).

Few series with different aims and methodologies have assessed the simultaneous occurrence of MM in patients with AL. Results similar to ours were reported by an early study from Mayo Clinic [6] in which 20 out of 147 pts (14%) had concurrent MM. This was a retrospective study in pts recruited at the end of the past century and selected on a random basis to measure circulating peripheral blood PCs. Only 9 pts (6%) underwent ASCT, heart involvement was present only in 38% of pts, and the median OS of the cohort was 25 m. In a larger series of 1255 AL pts (evaluated at Mayo Clinic within 90 d of diagnosis) [7], the authors found 8% of pts with MM, by the presence of CRAB, with median OS of 10.6 m vs. 29 m for the rest of AL. They also showed that pts with >10% BMPCs without CRAB had a poor prognosis, similar to pts with CRAB. 28.9% of the cohort underwent ASCT. Other small series showed an unexpectedly higher incidence of MM/AL [8] but these results were considered controversial. A recent population-based series from Canada with 215 pts identified “concurrent” MM in 29.8%. However, the definition of concurrent was any MM diagnosis recorded in the medical notes or reported to the provincial cancer registry occurring within 12 months prior to the date of diagnosis with AL amyloidosis or anytime during the follow-up period post-AL diagnosis. Therefore, practically all the scenarios were included. Moreover, a claims-based algorithm was used to identify cases with the risk of misclassification. Cardiac involvement was present in 67.9%. Localized AL amyloidosis was not specifically ruled out. Median OS was 39.9 m, 95% CI, 25.6–67, from the time of diagnosis [9]. A retrospective and prospective study from Korea with 302 AL pts found 59 (19.5%) with simultaneous MM criteria. Globally, the heart was involved in 68.9% and 28.1% died early within 6m; median OS was 42 m, 95% CI 29.8–54.2. Both the frequency of MM/AL and median OS are similar to our results [10]. A very recent single center, retrospective, consecutive series of 142 AL pts from China, with a median follow-up of 21 m, identified 62 (43.8%) of them with concurrent MM according to IMWG criteria. The study focused on cytogenetic abnormalities. Heart involvement was only 35.5% in MM/AL and 40% in AL. 9.7% pts of the MM/AL group underwent ASCT [11].

Other series evaluated the presence of AL in pts with MM, with heterogeneous results. A single center of 201 MM pts from Mexico in which fat-pad biopsy was done in all cases, showed AL in 68 (33.8%), 16 of them with and 52 without symptoms. Median OS for patients who had AL was 13 m compared with 64 m for those without AL (*p* < 0.004). The authors concluded that periumbilical fat-pad biopsy should be mandatory in all NDMM pts [12]. By 2011, 1180 pts with MM (959) or SMM (221) were seen at Mayo Clinic, having available BM biopsies in 144 (77 MM and 67 SMM). Congo red staining of the BM was not routinely done unless there is clinical suspicion of AL. AL was retrospectively detected in only 2 cases (1.38%) [13]. In a single center series of 70 MM pts from Poland [14], 33 NDMM and 37 relapsed or refractory MM AL was diagnosed on abdominal adipose biopsy in 18 (25.7%). No clinical signs of AL were demonstrated in many pts. Therefore, the authors stated that abdominal fat biopsy should be considered in every MM patient. A single center, retrospective, population-based study in Spain was updated with information about AL comorbidity in 303 MM pts, demonstrating concurrent symptomatic AL in only 7 (2.3%) [15]. A recent single center retrospective series of 158 MM pts from China (36 SMM and 122 MM), showed AL in 49 (31%) demonstrating that AL was an independent prognostic factor [16].

Overall, the evidence on the simultaneous occurrence of AL and MM is scarce and heterogeneous. A biased selection could be present in referral centers in comparison with population-based studies. On the other hand, the presence of one of these entities in clinical trials focused on the other disease is commonly an exclusion criterion. Therefore, no information is available from clinical trials. As expected, in our series the presence of MM in AL pts behaves as a poor prognosis factor, with a difference of more than 17 m in the median OS with respect to the group of pts without MM. However, this contrast did not reach statistical significance, probably due to the relatively small sample size of the MM/AL subgroup and the inclusion of SMM. Demonstrating the concurrence of AL/MM is of paramount importance from a clinical and epidemiological point of view. The approved standard of care and eligibility criteria for autologous transplants are different for each disease. The prognostic impact in terms of survival is very heterogeneous and particularly poor for the advanced cardiac stage in AL amyloidosis. Cardiac involvement is associated with frailty and intensive care. The incidence, prevalence, survival and mortality of both diseases are also distinct for each entity. Therefore, red flags for AL amyloidosis, particularly for cardiac involvement, should always be kept in mind [22].

This is a large series of pts diagnosed with AL amyloidosis enrolled in a prospectively maintained registry in a harmoniously developed multidisciplinary unit. Most pts were referred through the Cardiology service. To the best of our knowledge, our series is the largest Spanish single center series of AL pts so far. It is also the series with the highest percentage of heart involvement (90.3%). Despite this, our median OS of 45.9 m compares well with the above referenced studies, and it is similar to the median OS of 48.8 m documented in the EMN23 study, the largest real-world epidemiological study to date [23], showing data of 4480 AL pts across Europe, with heart involvement in 67.9%. Our study also has some limitations. As is usual in referral centers, a selection bias cannot be ruled out. Data for some variables are lacking in some pts, particularly in early years. The imaging tests used across the study have changed over time. Since 2020 we use high sensitivity (cTnIhs) instead of cTnI and its correlation with other types of troponins for staging remains to be determined.

Reaching a timely and efficient diagnosis, particularly for AL amyloidosis remains an unmet clinical need. The creation of multidisciplinary units that include the primary care team could improve the diagnostic delay. Clinical scenarios suggesting cardiac AL amyloidosis should be known and timely addressed by every health professional involved in the care of these pts [24]. Every effort should be made to standardize a risk-based screening approach for AL in the monitoring of every MG patient, particularly based on NT-proBNP, renal function and alkaline phosphatase. Patient associations have a critical role in expanding the knowledge and visibility of this rare disease among stakeholders and the whole society. In the MM/AL setting, AL is the main determinant of both outcome and management approach. Large prospective observational studies and specific clinical trials for pts with AL and MM are encouraged, to better understand the characteristics and prognostic impact of pts with this association.

## 5. Conclusions

AL amyloidosis remains an incurable and devastating disease, with a still excessive diagnostic delay. Median OS is approaching five years in the last decade, despite an almost universal heart involvement. MM is a complex and heterogeneous disease with variable outcomes. The association of both entities is not well characterized. Diagnosis should be based on current international consensus criteria. The most frequent type of association is the synchronous occurrence of both diseases. In our series of AL pts, 18.7% fulfill the diagnostic criteria of MM, most of them with a synchronous pattern. The prognosis of AL is dominated by the heart involvement. Our Cox regression model shows that NT-proBNP ≥ 8500 pg/mL is an independent prognostic factor for OS, whereas LVEF has a marginal effect. Besides, age is also a very strong prognostic factor for both AL and MM. Finally, ASCT is still a goal for AL pts. 21.6% underwent ASCT in our series and the great advantage in this subgroup is confirmed in terms of OS, behaving as an independent prognostic factor.

## Figures and Tables

**Figure 1 life-13-01518-f001:**
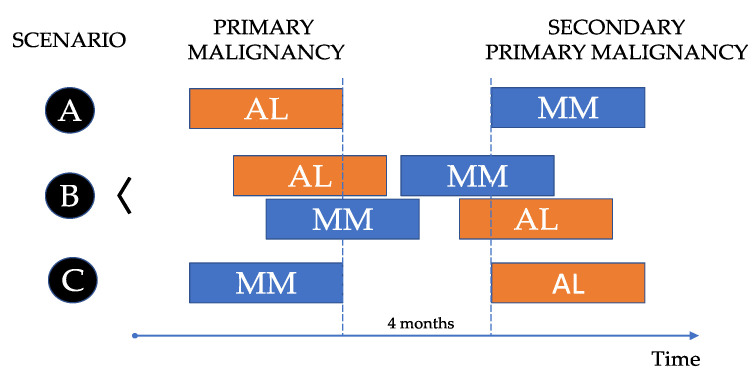
Scenarios on the association between multiple myeloma and systemic light chain (AL) amyloidosis.

**Figure 2 life-13-01518-f002:**
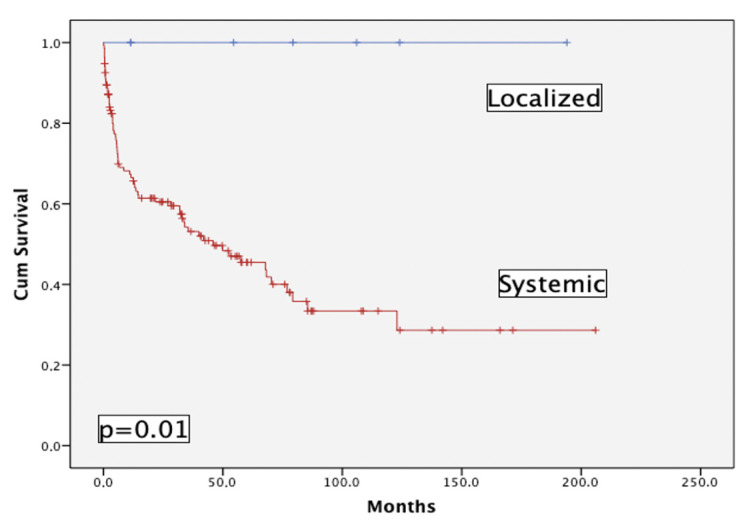
Overall survival of localized (n = 7) versus systemic (n = 134) AL patients.

**Figure 3 life-13-01518-f003:**
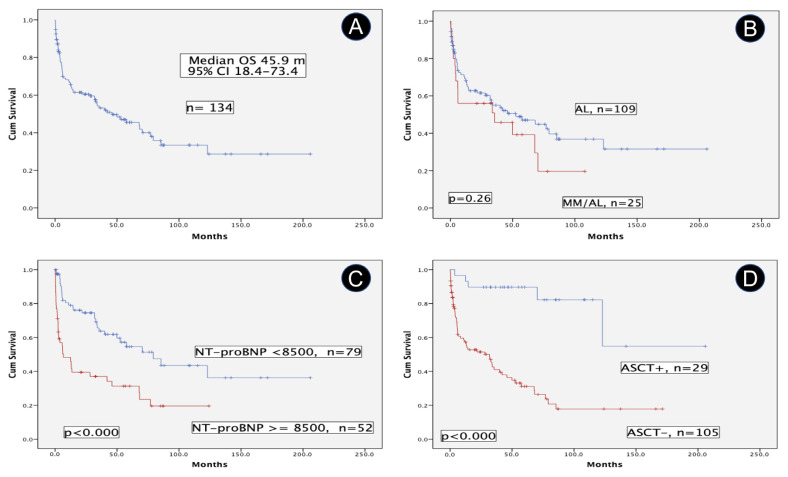
Overall survival in patients with systemic light-chain (AL) amyloidosis. (**A**) Overall Survival of the full cohort of systemic light chain (AL) amyloidosis. (**B**) Comparative overall survival of patients with or without multiple myeloma. (**C**) Prognostic impact of NT-proBNP (at cutoff of 8500 pg/mL) in terms of overall survival. (**D**) Comparative overall survival of patients who underwent autologous stem cell transplant versus non-candidates. Abbreviations: AL: systemic light-chain (AL) amyloidosis; ASCT: autologous stem cell transplant; MM: multiple myeloma; NT-proBNP: N-terminal pro-brain natriuretic peptide; OS: overall survival.

**Table 1 life-13-01518-t001:** Recent studies focusing on the association of multiple myeloma and systemic light-chain (AL) amyloidosis.

Author/Year	Period	Primary Disease	N	n (%N)
Pardadani, A. et al., 2003 [6]	1991–1996	AL	147	20 MM (**14**)
Kourelis, T.V. et al., 2013 [7]	2000–2010	AL	1255	100 MM (**8**)
Dinner, S. et al., 2013 [8]	2005–2011	AL	46	21/37 MM (**57**)
Jimenez-Zepeda, V. et al., 2022 [9]	2010–2019	AL	215	64 MM (**29.8**)
Yoon. S.E. et al., 2022 [10]	1995–2018	AL	302	59 MM (**19.5**)
He, H. et al., 2023 [11]	2012–2021	AL	142	62 MM (**43.7**)
Vela-Ojeda, J. et al., 2009 [12]	1989–2000	MM	201	68 AL (**33.8**)
Siragusa, S. et al., 2011 [13]	1993–2003	MM/SMM	144	2 * AL (**1.4**)
Usnarka-Zubkiewicz, L. et al., 2014 [14]	ND	MM	70	18 AL (**25.7**)
Ríos-Tamayo, R. et al., 2015 [15]	1985–2014	MM	303	7 AL (**2.3**)
Xu, J. et al., 2021 [16]	2010–2018	MM/SMM	158	49 AL (**31**)

Abbreviations: AL: systemic AL amyloidosis; MM: multiple myeloma; N: number of patients with de primary disease; n: number of patients with the associated alternative disease; ND: not determined; SMM: smoldering MM; *: retrospective study performed in bone marrow biopsies.

**Table 2 life-13-01518-t002:** Characteristics of patients at diagnosis.

Variable	MM/AL (n = 25)	AL (n = 109)	*p* Value
Age, years			
mean (SD)	61.60 (13.11)	64.94 (11.61)	ns
median (IQR)	61 (53–72)	65 (55–74)	-
Sex, % men	56	53.2	ns
ECOG 3–4, %	48	34.9	ns
Weight loss, %	40	30.3	ns
Diagnostic delay, mean (SD), months	6.58 (3.92)	8.02 (4.59)	ns
M-protein s mean (SD), g/dL	0.52 (0.82)	0.39 (0.65)	ns
M-protein s Biclonal, %	16	8.3	ns
FLCr (i/u) mean (SD)	173.58 (160.82)	32.78 (78.64)	**0.000**
FLCd mean (SD)	1337.47 (1211.73)	510.76 (1075.63)	**0.002**
Creatinine s mean (SD), mg/dL	1.01 (0.28)	1.32 (1.25)	**0.029**
24-h urine proteinuria mean (SD), g	3.20 (6.60)	2.87 (4.11)	ns
Heart involvement, %	92	89.9	ns
NT-proBNP mean (SD), pg/mL	9912 (16,974.82)	8811 (10,927.64)	ns
Troponin I mean (SD), ng/L	0.10 (0.07)	0.31 (0.83)	ns
LVEF %, mean (SD),	51.69 (12.77)	56.03 (12.27)	ns
Mayo 2012, IV, %	44	40.4	ns
Mayo 2004 mod. 2015, IIIb, %	28	26.6	ns
BMPC, mean %	37.16	18.02	**0.000**
BMI, mean (SD), Kg/m^2^	24.15 (3.02)	25.60 (3.86)	0.057
Num.comorbidities, mean (SD)	2.33 (1.74)	2.74 (1.90)	ns
Num.organs, mean (SD)	2.56 (1.29)	2.57 (1.11)	ns
Immunoparesis, %	95	76.7	0.064
Macroglossia, %	28	22.9	ns
CTS, %	28	14.7	ns
Prior MGUS, %	8.3	9.4	ns
ASCT, %	7/25 (28%)	22/109 (20.2%)	ns
Prior cancer, %	20.8	11.4	ns

Abbreviations: AL: systemic light-chain (AL) amyloidosis; ASCT: autologous stem cell transplant; BMI: body mass index (Kg/m^2^); BMPC: bone marrow plasma cells; CTS: carpal tunnel syndrome; ECOG: Eastern Cooperative Oncology Group; FLCd: free light chain difference; FLCr: free light chain ratio; IQR: interquartile range; LVEF: left ventricular ejection fraction; MGUS: monoclonal gammopathy of uncertain significance; MM: multiple myeloma; ns: not significant; NT-proBNP: N-terminal pro-brain natriuretic peptide; s: serum; SD: standard deviation.

**Table 3 life-13-01518-t003:** Univariate analysis and Cox regression model.

	Univariate Analysis			Cox Regression Model		
Variables	HR	95% CI	*p*	HR	95% CI	*p*
Age, years	1.05	1.03–1.08	0.000	1.05	1.02–1.08	0.001
ECOG 0–2 (vs. 3–4)	0.53	0.33–0.85	0.008	-		
NT-proBNP < 8500 (vs. ≥8500)	2.63	1.63–4.25	0.000	2.09	1.25–3.51	0.005
LVEF, %	0.98	0.96–0.99	0.036	0.98	0.96–1.00	0.06
BMPC, %	1.01	1.00–1.03	0.029	-		
Number of comorbidities	1.23	1.07–1.40	0.002	-		
ASCT	0.15	0.06–0.37	0.000	0.30	0.10–0.88	0.029
MM	1.38	0.78–2.42	0.265	-		

Abbreviations: ASCT: autologous stem cell transplant; BMPC: bone marrow plasma cells; CI: confidence interval; ECOG: Eastern Cooperative Oncology Group; HR: hazard ratio; LVEF: left ventricular ejection fraction; MM: multiple myeloma; NT-proBNP: N-terminal pro-brain natriuretic peptide.

## Data Availability

Not applicable.

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
