# Peer review of "AL Amyloidosis and Multiple Myeloma: A Complex Scenario in Which Cardiac Involvement Remains the Key Prognostic Factor"

_life, 2023, doi:10.3390/life13071518_

Round 1
Reviewer 1 Report (Previous Reviewer 3)
The authors sufficiently corrected the manuscript.
Author Response
Thanks to the reviewer
Reviewer 2 Report (Previous Reviewer 2)
The authors need to review the English language again. Some sentences do not follow on from previous ones and do not make sense.
Example in the abstract:
“ Monoclonal gammopathies (MGs) are a wide range of diseases which may evolve or pro-25 gress over time. Besides this, the co-occurrence of two entities is not a rare event.”
Which two co-occurrences?
Introduction:
- Lines 56-63
“AL amyloidosis is a rare and challenging disease, characterized by the presence of a 56 PC and/or a B-cell clone in the bone marrow (BM). ….. both diseases can be shown in the same patient at the same time or at 64 different times. “
Please review the definitions of both diseases; these are not scientifically accurate and the appropriate terms are missing.
- The term secondary primary malignancy does not apply in the case of MM and AL, please use an alternative term.
Results:
Table 1:
4th column: what does the n number refer to: AL? MM ?
5th column: what does the n and % number refer to? AL ? MM?
Please define
Table 2:
Cretinine ? do you mean creatinine
In terms of ASCT, which patients were eligible ? which patients were MM and which ones MM/AL only?
What are the characteristics of MM patients that are associated with a synchronous or metachronous development to AL?
Have you looked at that?
It would be interested to have these data?
Eg. λLC? Urine protein? Etc
Discussion:
I believe a point should be added about the value of identifying patients with MM who are at high risk for the development of AL (based on specific clinical characteristics ) and the need to monitor and screen at regular intervals for the development of AL and target organ damage. It is essential that clinicians look out for symptoms and signs that could point to the development of AL.
Also need to make clear in terms of synchronous AL/MM that it is AL that is the main determinant of both outcome and management and treatment approach. Patients with AL and MM should have a different treatment approach to MM alone and AL alone patients.
The authors need to review the English language again. Some sentences do not follow on from previous ones and do not make sense.
Example in the abstract:
“ Monoclonal gammopathies (MGs) are a wide range of diseases which may evolve or pro-25 gress over time. Besides this, the co-occurrence of two entities is not a rare event.”
Which two co-occurrences?
Introduction:
- Lines 56-63
“AL amyloidosis is a rare and challenging disease, characterized by the presence of a 56 PC and/or a B-cell clone in the bone marrow (BM). ….. both diseases can be shown in the same patient at the same time or at 64 different times. “
Please review the definitions of both diseases; these are not scientifically accurate and the appropriate terms are missing.
- The term secondary primary malignancy does not apply in the case of MM and AL, please use an alternative term.
Results:
Table 1:
4th column: what does the n number refer to: AL? MM ?
5th column: what does the n and % number refer to? AL ? MM?
Please define
Table 2:
Cretinine ? do you mean creatinine
In terms of ASCT, which patients were eligible ? which patients were MM and which ones MM/AL only?
What are the characteristics of MM patients that are associated with a synchronous or metachronous development to AL?
Have you looked at that?
It would be interested to have these data?
Eg. λLC? Urine protein? Etc
Discussion:
I believe a point should be added about the value of identifying patients with MM who are at high risk for the development of AL (based on specific clinical characteristics ) and the need to monitor and screen at regular intervals for the development of AL and target organ damage. It is essential that clinicians look out for symptoms and signs that could point to the development of AL.
Also need to make clear in terms of synchronous AL/MM that it is AL that is the main determinant of both outcome and management and treatment approach. Patients with AL and MM should have a different treatment approach to MM alone and AL alone patients.
Author Response
Thanks to the reviewer

This manuscript is a resubmission of an earlier submission. The following is a list of the peer review reports and author responses from that submission.
Round 1
Reviewer 1 Report
The authors described interesting status with AL and myeloma of large number of patients. However, there are some issues in this study.
1. The duration of enrollment is too long that have bias on the diagnosis and treatment courses.
2. The treatment modality and response should be considered in the outcome analysis.
3. For the risk of myeloma, chromosome and molecular study should be taken into consideration.
Author Response
Thank you for the comments.

Reviewer 2 Report
The authors present in a comprehensive and organized manner data from their center with regards to the co-existence and concurrent or sequential diagnosis of MM and AL amyloidosis.
I would recommend a few things
- Can the authors explain why they think the difference in OS between patients with AL only and AL/MM is not statistically significant? It is well established that outcomes are much poorer in patients with concurrent AL/MM and it is interesting that the difference in this cohort is not statistically significant. Any differences in timing/year of diagnosis or treatment received? Any other explanation?
- I would recommend reviewing and providing more data on the characteristics of MM patients who are more likely to develop AL amyloidosis. Do the authors have these data available in their patient cohort. Are there specific characteristics that make a MM patient more likely to have or develop subsequently AL amyloidosis?
- I would also recommend that the authors discuss what are the features the features that should raise suspicion and prompt the clinician to look for or suspect AL amyloidosis in MM patient. The pattern of proteinuria, type of FLC involvement, symptoms of heart failure? Should we screen specific patients?
- Please also discuss why it is important to diagnose patients that have concurrent AL/MM and how they should be treated differently
Author Response
Thank you for the comments.

Reviewer 3 Report
The authors analyzed the prognosis of
AL amyloidosis retrospectively. I have some comments.
1. In the simple summary, the contents seem to be apart from the content of the manuscript.
2. Introduction, the authors described about three scenarios, but the main results of the manuscript are largely irrelevant.
3. In line 235, the scenarios are firstly described. The authors should describe in the result section.
4. In table1, BMPC was 18.02% in AL amyloidosis. I think most of these patients should be diagnosed as myeloma.
5. In table1, LVEF has not shown in both groups.
6. In line 194, LVFE→ LVEF
7. Table2, the hazard ration of NT-proBNP is 1.00, meaning that NT-pro BNP is not associated with prognosis. But, the authors described NT-proBNP is a prognostic factor.
8. Figure3., I hope a more high-resolution figure.
Author Response
Thank you for the comments.

Round 2
Reviewer 3 Report
The authors adequately amended the manuscript.
Author Response
We appreciate the reviewer's kind comments